

# Understanding the progress of COVID-19 transmission in a rural district: a social network approach

Juliana Mansor[1,2], Nazarudin Safian[2], Fatimah Abdul Razak[3,4], Halim Ismail[2], Muhammad Haikal Ghazali[5] and Noriah Ismail[6]

[1] Pejabat Kesihatan Lembah Pantai, Jabatan Kesihatan Wilayah Persekutuan Kuala Lumpur dan Putrajaya, Kuala Lumpur, Wilayah Persekutuan Kuala Lumpur, Malaysia
[2] Department of Public Health Medicine, Faculty of Medicine, Universiti Kebangsaan Malaysia, Cheras, Kuala Lumpur, Malaysia
[3] Department of Mathematical Sciences, Faculty of Science and Technology, Universiti Kebangsaan Malaysia, Bangi, Selangor, Malaysia
[4] Centre for Modelling and Data Analysis (DELTA), Faculty of Science and Technology, Universiti Kebangsaan Malaysia, Bangi, Selangor, Malaysia
[5] Surveillance and Preparedness Unit, Department of Public Health, Selangor State Health Department, Shah Alam, Selangor, Malaysia
[6] Public Health Development Unit, Department of Public Health, Selangor State Health Department, Shah Alam, Selangor, Malaysia

Corresponding author
Nazarudin Safian,
nazarudin@ppukm.ukm.edu.my

## ABSTRACT

**Background:** Social interactions within and between communities influenced the spread of COVID-19. By using social network analysis (SNA), we aimed to understand the effect of social interaction on the spread of disease in a rural district.
**Method:** A retrospective record review study using positive COVID-19 cases and contact-tracing data from an area in Malaysia was performed and analysed using the SNA method through R software and visualised by Gephi software. The justification for utilizing SNA is its capability to pinpoint the individuals with the highest impact and accountability for the transmission of COVID-19 within the area, as determined through SNA.
**Result:** Analysis revealed 76 (4.5%) people tested positive for COVID-19 from 1,683 people, with 51 (67.1%) of the positive ones being male. Outdegrees for 38 positive people were between 1 and 12, while 41 people had 1–13 indegree. Older males have a higher outdegree, while younger females have a higher outdegree than other age groups among same-sex groups. Betweenness was between 0.09 and 34.5 for 15 people. We identified 15 people as super-spreaders from the 42 communities detected.
**Conclusion:** Women play a major role in bridging COVID-19 transmission, while older men may transmit COVID-19 through direct connections. Thus, health education on face mask usage and hand hygiene is important for both groups. Working women should be given priority for the work-from-home policy compared to others. A large gathering should not be allowed to operate, or if needed, with strict adherence to specific standard operating procedures, as it contributes to the spread of COVID-19 in the district. The SNA allows the identification of key personnel within the network. Therefore, SNA can help healthcare authorities recognise evolving clusters and identify potential super-spreaders; hence, precise and timely action can be taken to prevent further spread of the disease.

## INTRODUCTION

The COVID-19 pandemic persists as a global burden until today (*The Lancet, 2023*; *El-Sadr, Vasan & El-Mohandes, 2023*). Waves of COVID-19 infection occur globally and locally within countries. Although some countries have shifted their prevention and control actions towards an endemic state, some countries or areas still implement strict measures as part of their prevention and control programmes (*Tabari et al., 2020*; *Assefa et al., 2022*).

Researchers have demonstrated that COVID-19 is transmitted through droplets and direct contact, making a close encounter between a person infected with SARS-CoV-2 and a healthy person more likely to result in transmission (*He et al., 2020*). Given the mode of transmission of the viruses, studying the social relationships between people is crucial to understanding the virus and disease spread (*Sandeepa et al., 2020*; *Long et al., 2022*).

Social network analysis (SNA) is a technique which allows the analysis of infectious diseases' spread (*Herrmann & Schwartz, 2020*; *Manzo, 2020*). SNA builds its analysis on the social relationships among individuals or other social units (*Saraswathi et al., 2020*) and then generates human interaction network (*Razak & Zamzuri, 2021*). SNA uses the knowledge of the heterogeneity of the population in a real-world situation, differing from traditional analysis, which assumes homogeneity of the population (*Razak & Zamzuri, 2021*). Previously, SNA has been used in HIV and syphilis studies (*Klovdahl et al., 1994*; *Bell, Atkinson & Carlson, 1999*; *Young et al., 2013*) and SARS (*Zheng et al., 2008*). Additionally, through SNA, communities in the network have been identified (*Radicchi et al., 2004*; *Choudhury et al., 2022*), and high-risk individuals were pointed out (*Bell, Atkinson & Carlson, 1999*; *Christley et al., 2005*).

Given the recent pandemic, researchers have explored the usage of SNA in studying various angles of the COVID-19 pandemic, including modelling the transmission (*Manzo, 2020*; *Razak & Zamzuri, 2021*) and public sentiment through machine learning (*Hung et al., 2020*). Nevertheless, social interaction and relationships differ between people and communities based on their environment and cultural preferences (*Moya et al., 2020*). It has been demonstrated that sociocultural factors affect how COVID-19 spreads and how the government responds to mitigate spread (*Dascalu, 2020*; *Saunders & Schwartz, 2021*). Interestingly, a study observed that a suburb area was more vulnerable and severe to COVID-19 spread if the road network is highly connected (*Uddin et al., 2022a*), depending on COVID-19 variants (*Uddin et al., 2022b*). In contrast, a distinct pattern was seen in densely populated areas, where the cases were more widespread (*Dalziel et al., 2018*; *Ganasegeran et al., 2021*).

Rural areas commonly have a lower population density than urban areas, resulting in a lower infection rate. However, it was found that population density does not relate to COVID-19 spread (*Reza Khavarian-Garmsir, Sharifi & Moradpour, 2021*). Therefore, the study's main aim is to determine and characterise the social network of COVID-19 spread

in a rural district of a densely populated state in Malaysia with the highest number of COVID-19 cases (*KKM, 2020*). Additionally, the district was recorded as one of the districts with the highest number of cases in Malaysia's early phase of COVID-19 and has been categorised as a red zone area (*Hamsuddin, 2020*). Hence, we want to further discover how COVID-19 spreads in this district. Furthermore, we aim to determine the influence of a person in spreading COVID-19 in the network during the early phase of the pandemic through the SNA method.

## MATERIALS AND METHODS

### Study area

Selangor, located in the middle of East Malaysia, consists of nine districts with a population of nearly seven million, accounting for 20% of the total Malaysian population (*Jabatan Perangkaan Malaysia, 2021*). Hulu Selangor is one of the rural districts in Selangor, located in the northern part of the state. See Fig. S1 for the map of Selangor. With a population of 243,029 people (3.5% of Selangor's population) and a density of 139 people per square kilometre (*Jabatan Perangkaan Malaysia, 2021*), Hulu Selangor is one of the rural districts with a high number of cases in the early phase of the pandemic compared to other rural districts in Selangor. The district has a mix of socioeconomic and sociocultural activities, which can influence the spread of COVID-19.

### Data source

Data were obtained from the Selangor State Health Department (SSHD) and Hulu Selangor District Health Office (HSDHO) from January 1, 2020, until August 31, 2020. The study period coincides with both the first and second waves of the COVID-19 pandemic in Malaysia (*Fook Chris Sheng et al., 2020*). During this time, a standard testing method, the nasopharyngeal swab, was used for the COVID-19 polymerase chain reaction (PCR) test for all people under investigation. The data gathered was mainly on the positive cases of COVID-19 and their respective close contacts from the existing database and investigation forms. The data included addresses for each patient and their related contacts, COVID-19 lab findings, the relationship between the cases and contacts, the status of COVID-19 symptoms, and sociodemographic variables like age, race, and citizenship. These variables are known as 'attributes' in SNA. The data gathered was made anonymous, with each case given a new ID number. All cases and their respective close contacts for the period were included in this study to visualise the network of COVID-19 infection in the community. A directed, unweighted network graph was constructed based on the data gathered. The graph represents the network of COVID-19 infection in the community. Imported cases were excluded from the analysis since they do not contribute to forming significant clusters.

### Study design and analysis

This study is a retrospective record-review study that uses SNA as part of its analysis method. Relevant data, such as demographic details and lab results, were extracted and tabulated. Descriptive analysis was performed using the *"dplyr"* (*Wickham et al., 2018*)

package in R software (*R Core Team, 2021*). The analysis was divided into two sections: Network A, which includes COVID-19 cases and their contacts regardless of their lab test result, and Network B, which includes cases and contacts with a positive lab test result. Subsequently, nodes and edge datasheets were constructed based on the available data. The analysis using SNA is highly dependent on the nodes and edge datasheets, which provide the foundation for the network analysis. Nodes correspond to each individual, regardless of their status as a case or a contact. Edges represent the connections between individuals, where cases and their respective contacts were matched accordingly. The edge's direction depends on whether the person was named as their close contact or if both nodes named each other as their close contact, resulting in bidirectional edges. In the node's datasheet, each individual's attributes were included.

Meanwhile, relationships between cases and contacts were included in the edge datasheet. Before the analysis, the node and edge datasheets were combined to form the network, which was then used for further analysis. To perform this, the "*igraph*" (*Csárdi & Nepusz, 2006*) and "*statnet*" (*Handcock et al., 2008*) packages from R software (R Studio version 1.4.17171) were used for network analysis and Gephi version 0.9.2 for visualisation. In Gephi, the Force Atlas two layout algorithm was used to visually represent clustered events for an extensive network (*Gephi, 2011*). The relationships between positive cases and their close contacts were examined using a one-mode network. The descriptive analysis and visualisation of the network were performed to get an overview of the whole network. To identify the most influential person in spreading infection in the network (*Christley et al., 2005*), three key centrality metrics were calculated: (i) degree (the number of persons connected to each person) (*Borgatti & Ofem, 2010*); (ii) betweenness (how frequently a node lies on the shortest path between two nodes) (*Saraswathi et al., 2020*) and harmonic closeness centrality (how efficiently a node transmits the infection) (*Rochat, 2009*) (Fig. S2 shows the calculation of the centralities). Influential nodes in this study referred to the node with a high degree and high betweenness centrality value. The definition of network terminology is described further in Article S1. We also calculated the modularity value, which measures the strength of the division of the network and helps identify communities (clusters) (*Ji et al., 2015*).

Furthermore, we also identified super-spreader nodes, which we define as any node with an outdegree ≥5. It refers to the person who infects five or more people as a "super-spreader agent" (*Adegboye & Elfaki, 2018*). In addition, the location of each person was also mapped based on the available address. Each address was geocoded into latitude and longitude using MyGeoTranslator version 2.0 (*National Geospatial Centre, 2020*) and then mapped. Subsequently, nodes were mapped using QGIS version 3.30 (*QGIS Developers, 2023*) software. By doing this, we could see the distribution of cases and how COVID-19 has spread in the area.

## The terminology used in the study

Commonly used terminology in this study is explained in Article S1.

## Ethical considerations

The study has obtained approval from the National Medical Research Registry of Malaysia (NMRR ID: NMRR-20-2850-57309 (IIR)) and the Research Ethics Committee (REC) of Universiti Kebangsaan Malaysia (Project code: FF-2021-069).

## RESULTS

We analysed 1,683 nodes comprised of COVID-19 cases and their respective close contacts, labelled as Network A in Table 1. The network comprises 860 (51.1%) male nodes, with the highest age group between 5 and 12 years. The highest nodal degree for Network A was 143 (Table 1). All COVID-19 cases were linked to at least one close contact, who could either have a positive or negative laboratory test. Meanwhile, Network B consists of 76 (4.5%) nodes, with the age group between 18 and 29 years being the highest among COVID-19 cases. Most of the COVID-19 cases involved Malaysian citizens and members of the Malay community (Table 1). Network B has 219 links connecting the nodes. The average degree of the network was 2.882. We found 38 nodes had zero outdegrees, while the remaining half had 1–12 outdegrees, with 12 nodes having the same highest outdegrees.

Meanwhile, 41 nodes have an indegree greater than one, reflecting that the source of infection for these nodes could be more than one person. We noted that 33 nodes have zero-degree centrality, implying isolated nodes within the network. The highest outdegree for male nodes was 12, while the maximum outdegree for females was four. The mean outdegree of the network was 2.9.

The 95$^{th}$ percentile cut-off value for outdegree is 12. There are 12 nodes with an outdegree ≥12. They accounted for 132 (60.27%) of 219 links. All these nodes were men. Taking nodes with five or more outdegrees, 15 super-spreaders nodes have been identified (See Fig. S4). There are 42 communities in the same network. The network density was 0.039 with a network diameter 3, and the clustering coefficient was 0.960. The network reciprocity was 0.940. The betweenness centrality ranged from 0.09 to 34.5 for 15 nodes (Table S1), while others were zero. Only 15 nodes are bridges, transmitting the infection between patients without direct contact.

The aggregate network graph, created with Gephi, contains nodes that represent patients. Many probable large outbreaks would have occurred if the virus transmission had not been halted, as many clusters have formed based on Network A. However, only one large outbreak has occurred, involving 13 nodes, as seen in Network B (Fig. 1). Meanwhile, in Fig. 2, the nodes are coloured according to gender, race, or citizenship and sized by their centrality values. The aggregate positive cases network (Network B) contained 42 communities, with only three communities made up of five or more nodes, which accounted for about one-third of all nodes (29%, 38.2%) and nine-tenths of the edges (199%, 90.9%) concentrated within them. Malay, male nodes have a high degree centrality, but unlike female nodes, which have the highest betweenness centrality. The males were more connected and closer to the other nodes, while one of the female nodes played the key role in bridging the transmission to another indirectly connected node (Fig. 2). Furthermore, male nodes were approximately 40 years old, with a similar distribution
**Table 1 Networks parameters for the full network and the positive cases network.** Data is presented in two big sections: descriptive parameters of the data and network characteristics, which are further subdivided into node and network attributes; n = number. Abbreviations: %, percentage; *missing value was excluded from calculation; [a]exclude non-citizen.

| Parameters | | Network A (Full network) n (%) | Network B (COVID-19 cases) n (%) |
|---|---|---|---|
| **Total subjects** | | **1,683** | **76** |
| Gender | Male | 860 (51.1) | 48 (63.2) |
| | Female | 823 (48.9) | 28 (36.8) |
| Citizenship | Malaysian | 1,646 (97.8) | 73 (96.1) |
| | Non-Malaysian | 37 (2.2) | 3 (3.9) |
| Race[a] | Malay | 1,405 (85.4) | 62 (84.9) |
| | Chinese | 69 (4.2) | 1 (1.4) |
| | Indian | 84 (5.1) | 2 (2.7) |
| | Others | 90 (5.4) | 8 (11.0) |
| Age group | Mean (S.D) | *28.03 (18.7) | *33.6 (18.4) |
| | 0–4 years old | 54 (3.2) | 1 (1.3) |
| | 5–12 years old | 361 (21.4) | 9 (11.8) |
| | 13–17 years old | 264 (15.7) | 8 (10.5) |
| | 18–29 years old | 258 (15.3) | 17 (22.4) |
| | 30–39 years old | 261 (15.5) | 14 (18.4) |
| | 40–49 years old | 171 (10.2) | 8 (10.5) |
| | 50–59 years old | 165 (9.8) | 11 (14.5) |
| | 60–69 years old | 63 (3.7) | 7 (9.3) |
| | 70–79 years old | 28 (1.7) | 1 (1.3) |
| | ≥80 years old | 13 (0.8) | 0 |
| | Missing data | 45 (2.7) | 0 |
| **Network characteristics** | | | |
| | | **n** | **n** |
| Nodes | | 1,683 | 76 |
| Edges | | 2,231 | 219 |
| **Node attributes** | | **Range (mean)** | **Range (mean)** |
| Outdegree | | 0–141 (1.326) | 0–12 (2.908) |
| Indegree | | 0–13 (1.325) | 0–13 (2.908) |
| Degree | | 1–143 (2.651) | 0–25 (5.815) |
| Betweenness | | 0.000–1,952.667 (4.146) | 0.000–34.500 (1.289) |
| Eccentricity | | 1–11 (5.810) | 0–3 (1.184) |
| **Network attributes** | | **Value** | |
| Diameter | | 4 | 3 |
| Radius | | 1 | 0 |
| Mean path length | | 2.016 | 1.338 |
| Density | | 0.001 | 0.039 |
| Clustering coefficient | | 0.043 | 0.960 |

**Note:**
*Calculation of mean does not include the missing value.

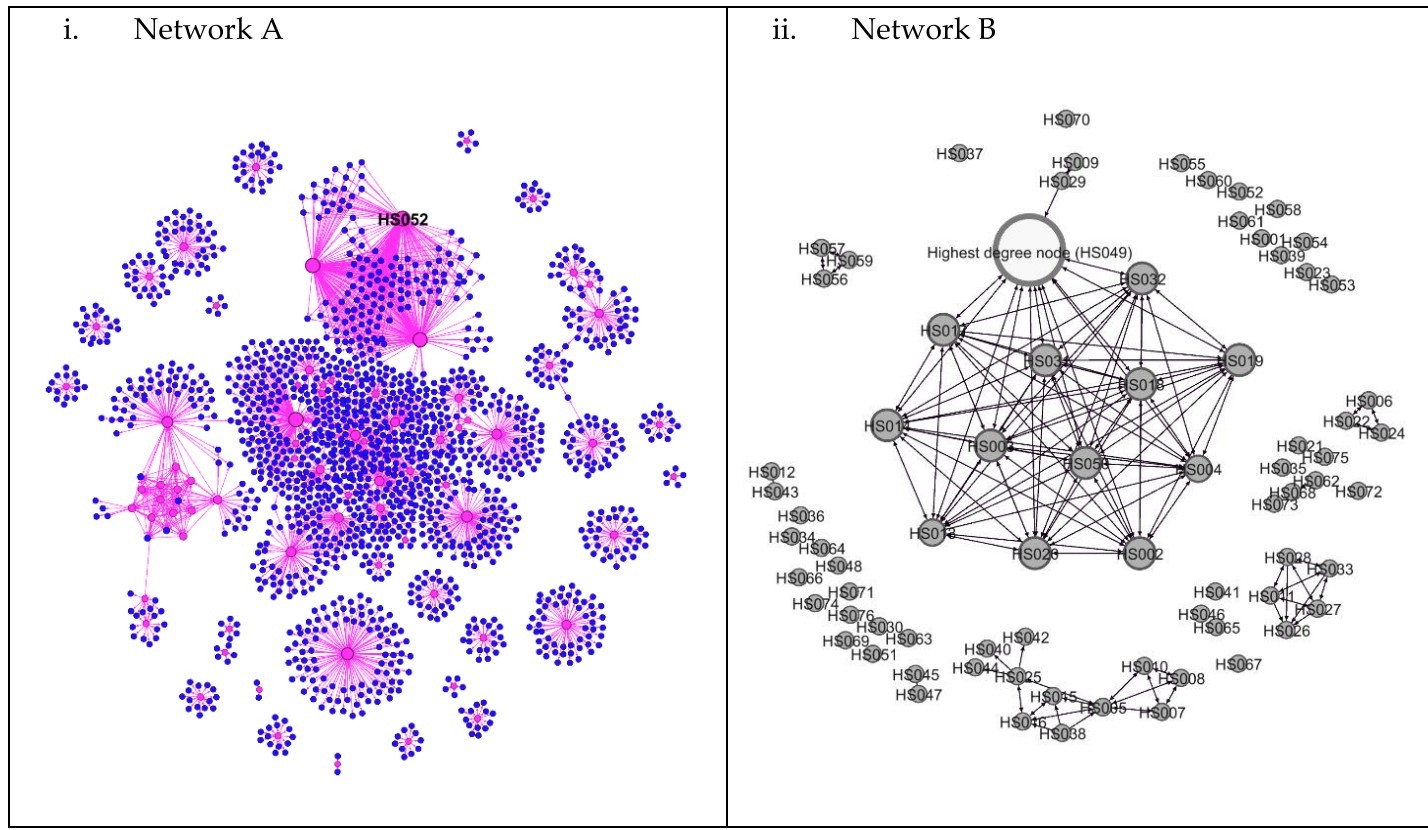

**Figure 1 Network of COVID-19 cases and its contacts.** (i) Network A. Aggregate network graphic created in Gephi. Node size is determined by degree; pink circles are positive COVID-19 cases, while the blue circles are negative nodes. Node HS052 is the node with highest degree (degree = 143). (ii) Network B. The white node is the highest degree node (degree = 25).

between the source and target age groups. Females are roughly 30 years old, with an inverse proportion of age between the source and target populations (Fig. S4). Figure 3 illustrates that male nodes exhibited greater involvement in Tablighi and educational clusters, with connections primarily established through social contacts during religious engagements. Conversely, female nodes, predominantly associated with educational institutions, displayed edges primarily linked through familial connections.

There were three large communities, two of which were from Tablighi, and one was an educational institution. The largest community has 13 (17.1%) nodes with 155 (70.8%) edges, while the second-largest community has 11 (14.5%) nodes with only 28 (12.8%) edges. The largest community mainly involved those who attended religious gatherings of the Tablighi (Fig. 3). All the nodes were male, with two being non-citizens. Meanwhile, the second-largest community involved family members and work-related close contacts. Non-connected nodes of the outbreak were also associated with religious events or places, such as mosques and religious institutions. Although they might not have been in direct contact or proximity with the person in the outbreak, perhaps there was some casual contact between the source and the infected person.

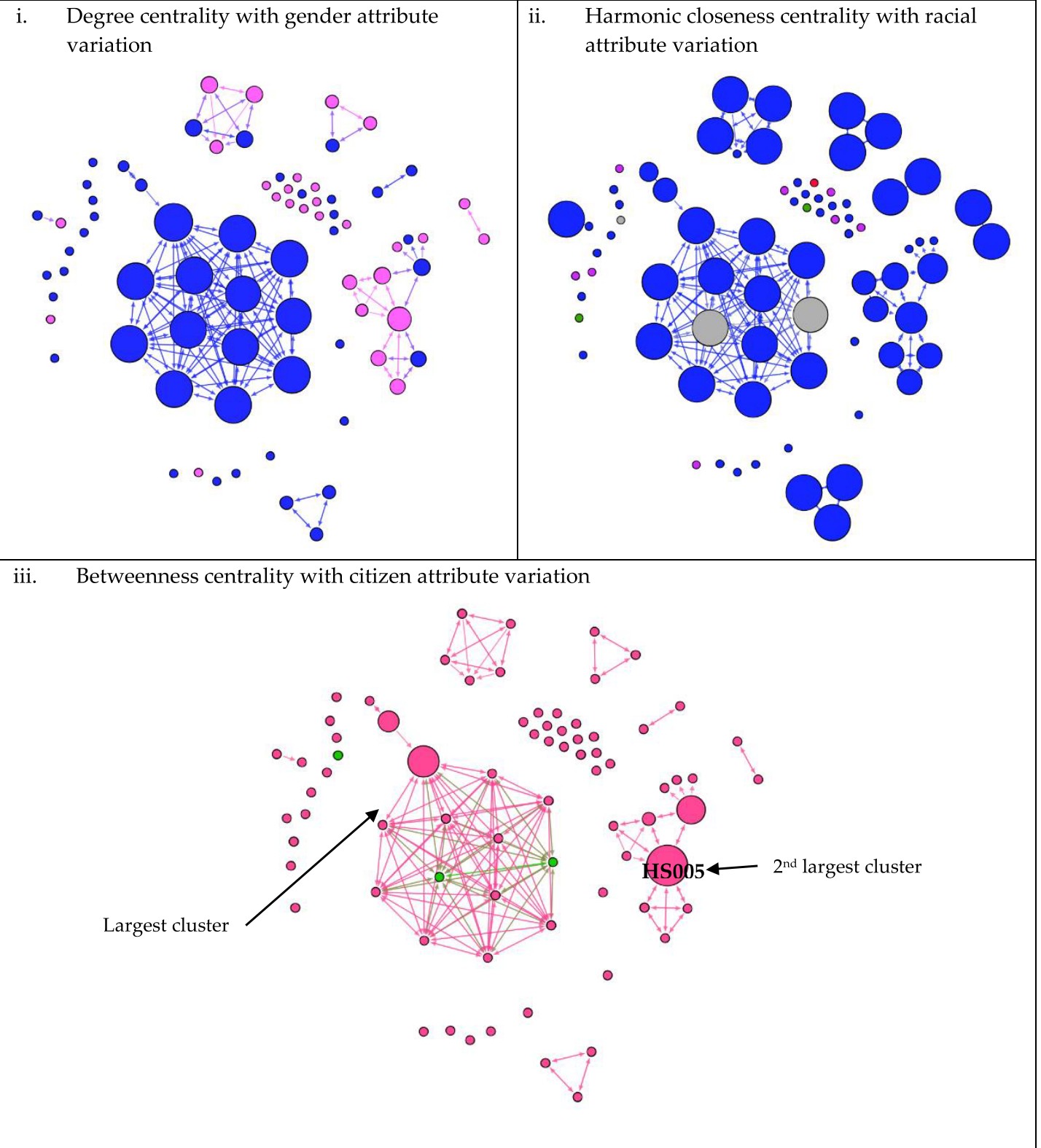

**Figure 2 Network of COVID-19 cases based on its centralities value and attributes.** Arrowheads indicate the direction of transmission from the source node to the target node. In Figures 2 (i) degree centrality, (ii) harmonic closeness centrality, and (iii) betweenness centrality, node size is ranked according to centrality value mentioned in each of the subfigure, increasing in size and value accordingly. Figure 2 (i): Males were represented by blue nodes, while females were represented by pink nodes. Figure 2 (ii): Blue-coloured nodes denoted Malays; purple-coloured nodes denoted

**Figure 2** (continued)
Orang Asli (Aboriginal); red nodes denoted Chinese; green nodes denoted Indian; and grey nodes denoted others (foreigners). Figure 2 (iii): Nodes coloured pink denoted Malaysians, while green nodes were non-citizens.

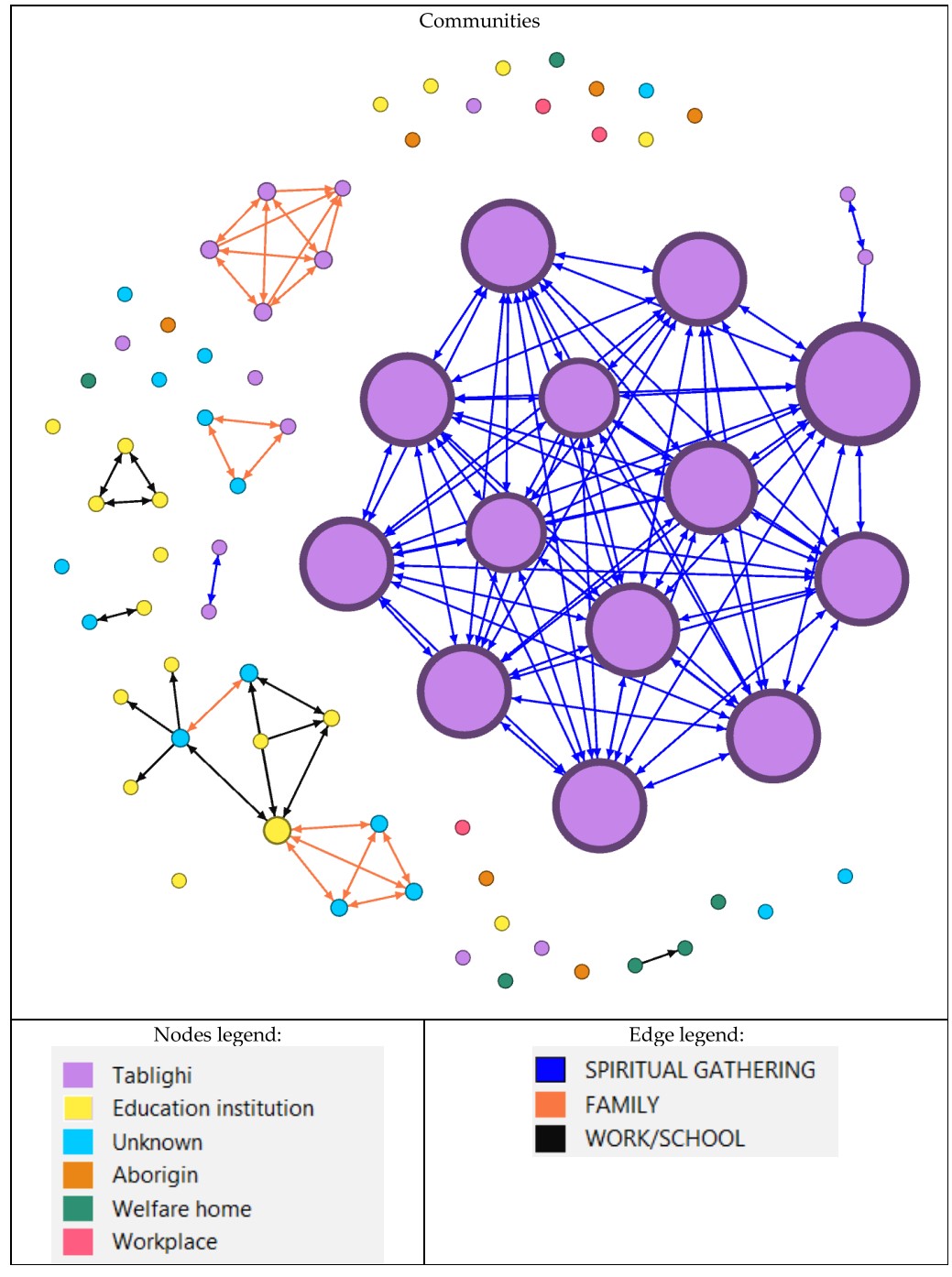

**Figure 3 Network of COVID-19 cases based on communities.** Groups of nodes are based on their community. Node size is determined by degree value, with the largest node having the highest value and vice versa. Nodes are classified according to the type of outbreak. An education institution includes a school or tertiary education centre. While the edge legend is based on the type of relationship between the nodes.

## Progression of the COVID-19 transmission

The first wave of COVID-19 in Malaysia started on January 25, 2020, and continues until February 16, 2020. During the first wave, no case was recorded in this district. Subsequently, the second wave started on February 27, 2020, where all the nodes were observed (Fig. 4). Although the first wave started in January 2020, total lockdown, also known as movement control order (MCO), was implemented only from March 18 until May 3, 2020. Subsequently, a conditional movement control order (CMCO) was issued until June 9, 2020, and a recovery movement control order (RMCO) was issued from June 10, 2020, until March 31, 2021. The infection network started with four initial nodes in the pre-lockdown period, which comprised one sporadic case and three related nodes. It then rapidly evolved into a more complex network during the MCO period. Most of the nodes were infected during this period. Although the declaration of MCO occurs early, the disease has spread earlier and has only been detected during the MCO period. During the CMCO period, the number of nodes decreased, and most cases involved sporadic cases. Lastly, only a single sporadic node was recorded during the RMCO period until the end of the second wave.

## Location of nodes

Based on Fig. 5, some nodes were located outside the territory of Hulu Selangor. It shows the infection might have been extended outside the district if the cases have travelled back to their hometowns or originated from the address given. All nodes outside the district were involved in the largest community, which originated from religious gatherings. Two of the nodes in the community (HS002 and HS003) were among the earliest cases recorded in the district and with addresses outside the district (Kuala Lumpur and Kelantan, respectively). Thus, the source of infection in this community might originate from outside the district. Meanwhile, inside the district, cases were more concentrated in the local town area, where the administrative and economic activity runs.

## DISCUSSION

We have visualised the social network based on contact tracing data to derive insights into the pattern of COVID-19 transmissions in this district. This study reveals that most COVID-19 cases in Hulu Selangor involved early- to mid-elderly males. The age and sex profile of our data corresponds with Malaysia's nationwide surveillance data, with age distribution and COVID-19 primary attack rate close to our study population (*Jayaraj et al., 2021*). Similar distributions were observed in a few studies in the United Kingdom and European countries (*Gebhard et al., 2020*; *Azizi et al., 2022*). The ratio between males and females and the age difference between the two groups infected with COVID-19 could be linked to social activity, lifestyle, behaviour, and comorbidities (*Gebhard et al., 2020*; *Azizi et al., 2022*). Chronic comorbidities are more common in males and have been linked to risky lifestyles and behaviours such as smoking and drinking (*Cai, 2020*). Although the characteristics were not investigated in this study, Malaysia's statistics have shown an increase in the prevalence of chronic comorbidities among Malaysian adults and the elderly (*Institute for Public Health (IPH) Malaysia, 2020*). Hence, similar conditions can be

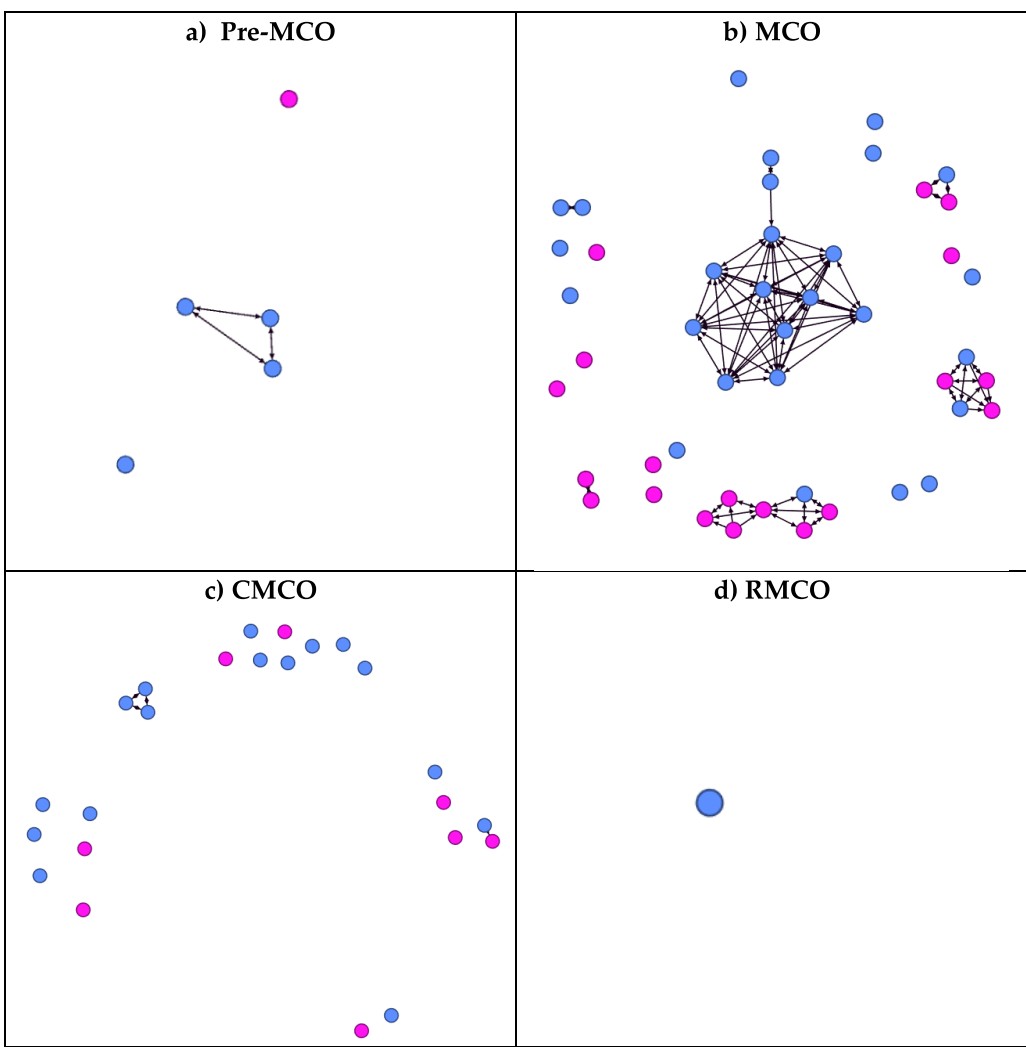

**Figure 4 Progression of the network according to the different phases of COVID-19 infection.** During the first wave of COVID-19 in Malaysia, no cases were recorded in the district. All cases in the district were documented during the 2nd wave of COVID-19. (A) During the pre-lockdown period. (B) During the Movement Control Order (MCO): 18 March 2020 until 3 May 2020. (C) During the Conditional Movement Control Order (CMCO): 3 May 2020 until 9 June 2020. (D) During Recovery Movement Control Order (RMCO): 10th June 2020 until 31st March 2021. The blue circle denoted male, while the pink circle denoted female.

postulated for the communities. Additionally, as most cases were linked to religious gatherings (*tablighs* and mosques), usually attended by a majority and regularly by adults and older men, thus the cases were mainly distributed among men (*Mat et al., 2020*).

Most cases involved in outbreaks are those linked to a single religious gathering. Malaysia has recorded a large outbreak of COVID-19 that is causing international spread due to a single mass gathering of a religious missionary movement called Tablighi that occurred at the end of February 2020. The gathering involved multiple countries, with activities such as sharing sleeping areas in confined spaces and being close to persons (*Mat et al., 2020*). Besides being in their groups' circles, they also tend to move from one place to

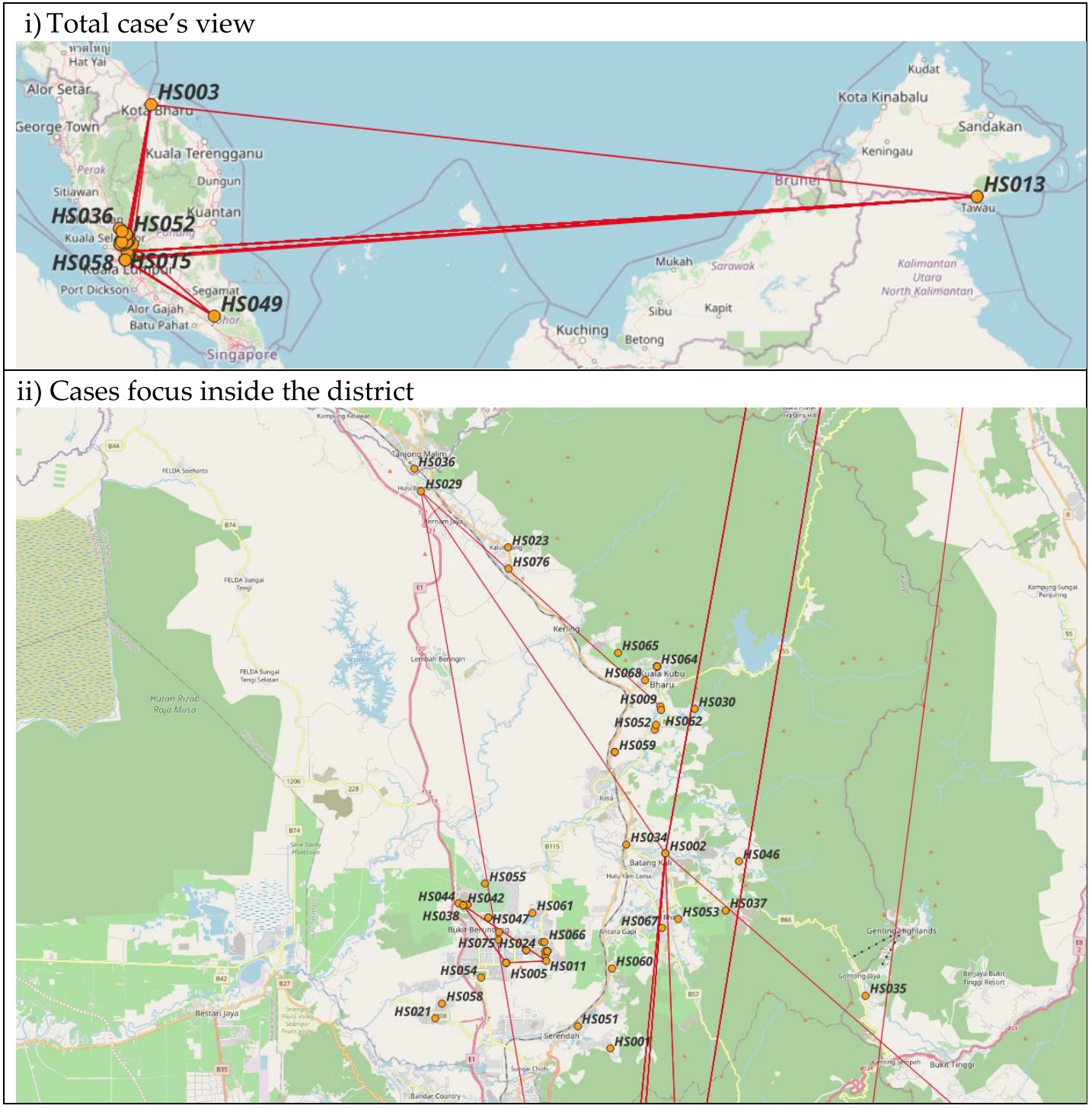

**Figure 5 Map of COVID-19 cases.** (i) and (ii): Nodes location based on address available. Maps by © osm-tools.org & OpenStreetMap contributors, CC-BY-SA.                                                                                     

another for their *dawah* and *tabligh* (proselytisation) (*Zulkifli, 2021*) activity, which includes mosques, religious institutions or schools, and local communities' houses. Because of this known activity, they might have met the local people, students and teachers

of religious institutions (*Hashim et al., 2021*). Therefore, the individuals may spread the infection unconsciously, as they do not know each other well, and the exact address or location cannot be determined or recalled during the health authority's investigation through manual contact tracing. Hence, the effect of sporadic cases or small clusters in religious institutions and the local community.

Given the large crowd and the proper registration of those who came for the gathering was not performed, hence it is difficult to trace people by manual contact tracing. Manual contact tracing is a tedious, time- and resource-consuming activity with limitations for the parties involved (*World Health Organization, 2021*). Using digital contact tracing applications and techniques can significantly help overcome these issues. Nevertheless, the digital contact tracing method also has challenges and issues, such as the user's adaptability, privacy and ethical issues, and technology limitations, such as connection with the internet or signal transmission and transparency (*Shahroz et al., 2021*).

The spread of the COVID-19 infection relies on many factors. High betweenness centrality of a node will affect the size of an outbreak, especially if the same node also has high HCC (*Vargas et al., 2018*). Removing the edges can reduce the outbreak's size (*Bergamini et al., 2018*). Hence, fewer people will be exposed and be at risk of getting infections. In this study, a female node bridges the nodes in the second large outbreak (Fig. 2). It relates to the fact that within a family, women tend to handle the house chores and family members (*Roberson & Gebeloff, 2020*; *Stokes & Patterson, 2020*). Given the nature of the virus transmission (*Manikandan, 2020*), it explains the nodes's high betweenness and HCC value. Therefore, working female caregivers in a multigenerational home have been found to spread the infection. They also work outside the home, where they are exposed to the infection and care for their households, spreading the infection to the children and elderly (*Stokes & Patterson, 2020*). Thus, having a proper policy and prevention programme for working females could prevent further spread of the disease (*Schulz, 2020*; *Stokes & Patterson, 2020*). Besides, the most crucial thing is ensuring that working females have good awareness and knowledge of the proper use of face masks and hand hygiene, especially at work (*Keleb et al., 2021*).

A movement restriction order was implemented to limit the contact between people, thus reducing the number of new and active COVID-19 cases (*Tang, 2022*). Nevertheless, the initial period of the lockdown phase observed a peak in the number of cases for the first 2 weeks as people moved across the country back to their hometowns and as healthcare authorities tried to catch up with the investigation and screening of cases (*Salim et al., 2020*). The pattern of COVID-19 evolution in our network coincides with the nationwide cases, also caused by the mass gathering of Tablighi (*Salim et al., 2020*; *Hashim et al., 2021*). The mass gathering occurred in the city centre of Kuala Lumpur, about 60 km from Hulu Selangor, on February 27, 2020 (*REUTERS, 2020*). It initiated the second wave of the outbreak for the whole country, but the first few cases were only detected about a week after the gathering had ended. The detection time gap allowed for the infection to spread elsewhere (*Hashim et al., 2021*). MCO was only implemented a week after the initial detection, which further caused the infection to spread. The mass gathering also triggered the authority to make a public announcement for the Tablighi members and anyone in

contact with them to come forward for screening, further increasing the catch of cases (*Abdullah, 2020*). Thus, it explains the non-connected nodes that are the source of infection from Tablighi.

A reduction in cases was observed only after a few weeks of MCO, similar to our findings (*Tang, 2022*). The reduction in cases allowed some restrictions to be lifted, resulting in sporadic cases and a small outbreak in the workplace and within families. Sporadic cases occur as contact tracing cannot identify their linkage to other outbreaks. Since COVID-19 viruses are airborne and droplet transmission, it is more difficult to determine the location of contacts as the virus can land on surfaces and be unconsciously picked up by other people, although the survivability of the virus in the environment is debatable (*CodeBlue, 2021*; *Ahmad & Pfordten, 2021*). Nevertheless, the spread of infection was contained as the community became more vigilant and adopted preventive behaviours such as wearing masks, practising good hygiene, using cough etiquette, and social distancing, and the health authority was able to do contact tracing, quarantine all close contacts and isolate cases (*Manikandan, 2020*; *Chiu et al., 2020*).

The $R_o$ in Malaysia during the first and second waves of COVID-19 was reported to range from less than one to around 2.2 just at the beginning of September 2020 (*Mok, 2020*), with an effective reproduction number, $R_t$ around 0.9 to 1.2 for the first and second waves (*Jayaraj et al., 2021*; *Musa et al., 2021*), slightly lower than the finding of our study outdegrees Nevertheless, one study by *Fook Chris Sheng et al. (2020)* reported $R_t$ at 3.1 during the peak of the second wave. Although the $R_o$ from traditional mathematical epidemiology is commonly used to predict of the disease's epidemiological severity, it does have its issues, such as being difficult to estimate and relying on rough assumptions (*Heffernan, Smith & Wahl, 2005*). The difference in $R_o$ value relies on the assumption of mass-action contact and an undirected network in traditional mathematical epidemiology (*Holme & Masuda, 2015*; *Allard et al., 2022*). However, the $R_o$ and size of the disease spread depend on direction (in- and out-degrees), heterogeneity of the population, and correlation (*Kao, 2010*; *Allard et al., 2022*).

Every study has its limitations, including this one. Above all, the data has its limitations because it is pre-collected. There were missing variables, also known as item non-response (*Huisman, 2009*), especially on the negative close contacts since, in the early phase of the outbreak, there were no standard forms of documentation, and it was entirely based on the district's efforts, rather than uniform documentation from the state or nationwide. For instance, the data form did not capture whether the individual was symptomatic or when symptoms started. Hence, we had to drop this variable from the analysis, which does not let us see the temporal aspects of the analyses.

On the other hand, we did include non-response items on outbreak types and relationships and labelled them as 'unknown'. Since our study just descriptively analysed the results and the missing values do not affect the relationship between nodes, we do not proceed with methods to treat the data, such as likelihood-based estimation and imputation (*Huisman, 2009*). In the future, a study using real-time, proper, and complete data collection could contribute to a deeper understanding of the issue. Nevertheless, our study has its strength in that the cases included were tested using the PCR method. PCR is

the gold-standard method for diagnosing COVID-19. Moreover, trained personnel collected and screened the data, making the contact network reliable. Besides, this study scientifically showed that external sources contributed to the spread of COVID-19 in suburban-rural areas, and the density of an area does not necessarily contribute to the spread of COVID-19 but rather the relations between individuals.

## CONCLUSIONS

The results from our study suggest that social relationships among individuals are essential in determining the magnitude and direction of COVID-19 spread. Women were important in bridging the infection, while older men were the popular nodes with many connections. Hence, educating both men and women on using face masks and additional strict hand hygiene for women is crucial. Working women caring for their small ones or older adults should be given priority for the work-from-home policy compared to others. Besides, any mass gathering could initiate a cluster and propagate the infection; thus, it should be prevented; if needed, it should be done with strict adherence to standard operating procedures. Therefore, SNA is essential to determine the pattern of social relationships and identify the potential super-spreader characteristics. In future, having real-time social network analysis software or an application in addition to the training of operators can significantly help them better understand, detect clusters, and control the outbreak timely, as well as predict and assess the outcome of any intervention conducted in the community.

## ACKNOWLEDGEMENTS

We appreciate the hard work of the ground healthcare worker who selflessly performed the contact tracing, which made the research possible. We also thank the Selangor State Health Department and district health offices in Selangor for their cooperation in providing the data and the Director General of Health Malaysia for permission to publish this article.

### Funding

This research was funded by Universiti Kebangsaan Malaysia Grant (GUP-2021-037 and GUP-2021-046). The funders had no role in study design, data collection and analysis, decision to publish, or preparation of the manuscript.

### Grant Disclosures

The following grant information was disclosed by the authors:
Universiti Kebangsaan Malaysia Grant: GUP-2021-037 and GUP-2021-046.

### Competing Interests

The authors declare that they have no competing interests.

## Author Contributions

- Juliana Mansor conceived and designed the experiments, performed the experiments, analyzed the data, prepared figures and/or tables, authored or reviewed drafts of the article, and approved the final draft.
- Nazarudin Safian conceived and designed the experiments, performed the experiments, authored or reviewed drafts of the article, and approved the final draft.
- Fatimah Abdul Razak conceived and designed the experiments, performed the experiments, authored or reviewed drafts of the article, and approved the final draft.
- Halim Ismail performed the experiments, authored or reviewed drafts of the article, and approved the final draft.
- Muhammad Haikal Ghazali performed the experiments, authored or reviewed drafts of the article, and approved the final draft.
- Noriah Ismail performed the experiments, authored or reviewed drafts of the article, and approved the final draft.

## Field Study Permissions

The following information was supplied relating to field study approvals (*i.e.*, approving body and any reference numbers):

Field experiments were approved by the National Medical Research Registry of Malaysia (NMRR ID: NMRR-20-2850-57309 (IIR))

## Data Availability

The attribute and link data for COVID-19 positive cases are available in the Supplemental Files.

## Supplemental Information

Supplemental information for this article can be found online at http://dx.doi.org/10.7717/peerj.18571#supplemental-information.

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
