# Peer review of "Understanding the progress of COVID-19 transmission in a rural district: a social network approach"

_PeerJ, doi:10.7717/peerj.18571_

## Round 0.1 · original submission · Major Revisions

Three independent reviewers all rated your manuscript positively. In the manuscript, very specialized terms associated with SNA field and associated Operations are used, and thus, the majority of reviewers indicated them as an essential point to be clarified and improved.

·

Basic reporting

1. The entire results section, discussion and tables+figures need reworking for technical and linguistic clarity.
The author switches between describing the larger overall network and the smaller 76 node network of positive cases, and it is difficult to make out which attributes belong to which network.
The results and discussion sections talk about differing sex, age and ethnic roles in driving spread of infection but there is no supporting table or figure for these findings.
The authors are advised to add tables/figures specific to the key result metrics so that readers will be able to understand. At present it is very difficult to understand what part of the network is being described. Detailed and lucid figures are vital as SNA is strongly associated with the science of data visualization.
2. Data availability:
a. There is no raw data/data availability statement for the larger network (which is also analysed and described in results), uploaded data file is only for the positive network of 76 nodes.
b. Some of the headings in the data file are not in English, hence difficult to understand.
3. Figures are not sufficiently high definition. They get pixellated if one tries to zoom in for detail.
4. The language of the manuscript should be improved to ensure that an international audience can clearly understand the text. The authors will benefit from proofreading and review done by a colleague who is proficient in English, or they may choose to seek professional editing services.

Experimental design

SNA field specific and Operational definitions need to be clearly laid out. The manuscript body should prominently reference the supplement file where definitions are listed.

Validity of the findings

Section-wise comments:
Abstract:
“The rationale for using SNA is that it allows for the identification of the most influential and responsible person for the COVID-19 spread in the area.”
The phrasing of this sentence makes comprehension difficult. Being familiar with SNA, I can grasp what the authors wish to convey, however other readers may interpret the words "responsible" and "influential" in social context instead of in terms of their centrality. I suggest you have a colleague who is proficient in English and familiar with the subject matter review your manuscript, or contact a professional editing service.

“Analysis revealed 4.5% positive nodes from 1683 total nodes, with 67.1% of the positive ones being male”
Please define “positive node” in this scenario and mention the absolute numbers along with the percentage.
“Older males have a higher outdegree, while younger females have a higher outdegree than other age groups. “
Overall or within their sex class?
Table 1:
Malay positive cases cannot be 629 if total nodes in positive network is only 76
Age group Mean (S.D) Why are the numbers starred?
Node attributes: Degree: column 1: For this number 143 to be possible, at least one node would have to have sum of indegree and outdegree equal to 143. Please include a figure showing the network position and contacts of the specific node/s and a corresponding figure to show the inward and outward degrees of the (positve case network) node that had degree of 25.

Figure 3: Node sizing according to which specific centrality measure, not clear from legend. Also, most of the edges are bidirectional. This needs further elaboration in results and discussion.
Figure 4: singular form of matrices is matrix.
Figure 5: Gray nodes not explained in legend

Line 79-80 Please specify the to and from properties of the edge unambiguously
Line 89 Pls provide operational definition of "influential"
Line 90-93 Each centrality measure definition needs citation
Line 93-94 It's preferable to use scientific terms instead of generic adjectives to describe node hierarchy
94-97 Please rephrase, since the role of modularity in identifying superspreaders is not clear
124 12 males, or 12 outdegree? Please clarify.
Line 125 “around 3” Pls write exact outdegree
126-130 Please add figures showing these specific nodes in the network
127 “components” Pls define
145-146 ?clarity. Ethnicity, sex and age segregated degree and betweenness metrics need to be tabulated and described.
146-148 In which network? Please reference with figure
148-150, 153-155 Needs clarity of phrasing and rewording for grammar and syntax
220-221 Please clarify the relationship between the preceding description of religious gatherings and manual contact tracing
231-232 Please show detail in network figure showing the nodes and edges of the 2nd outbreak

Reviewer 2 ·

Basic reporting

1) Clear unambiguous English was used throughout the manuscript
2) The introduction does show context but I would like to see more contextual background on the environment. Much has been published on social contact networks, which have been interpreted and analysed in numerous different ways (job, age, location etc.) so to be novel and of interest to the global community, the setting is key as differences can be identified as such in comparison to others. Please therefore provide substantial insight into the setting, why it is different to others and what you are trying to gain understanding in specifically in this location
3) The structure of the manuscript is appropriate
4) The quality of the figures (PPI) needs to be improved. Cannot read the text of Figure 6. Also there are too many figures. Some should go into the SI with a focus on the main.
5) I would assume that the raw data is unable to be shared due to the confidential nature of what has been collected

Experimental design

1) This is difficult to determine as this paper is more in the realm of social sciences and methodological, rather than biological or health related. The main goal is to ascertain contact networks rather than simulate outbreaks or public health impacts.
2) The research question is clearly defined in the fact that they wish to describe the social contact networks within a specific district. The evolution of the COVID-19 outbreak is a bit misleading as it is not biological or virological, rather the impact of NPIs on social contact networks.
3) More work I feel could be done on actually modelling the outbreaks or even a statistical analysis on the outbreak within this region. This could be done side by side with the social contact network analysis to estimate the R0. My question would be, what can we take forward as epidemiologists/public health modellers with the information/analysis carried out? Can you show that NPIs should be implemented differently in this particular rural setting versus urban settings in Malaysia?
4) The method is a standard SNA yet the interpretation should be cautioned – to know who infected who goes far beyond SNA and requires consideration of serial intervals. It cannot be necessarily inferred that one person infected the other outside of a black box setting based on social contact networks alone. Genetic analyses are required after the first 100 cases or so in isolation settings (ie within household, or imported cases entering a susceptible setting) as we simply cannot infer the weightage of the social network within an open environment in transmission dynamics. I also did not find as to whether information on the household/GPS location/workplace/place of worship etc. was obtained for the people in the network. This is not clear.

Validity of the findings

1) The benefits need to be more clearly outlined specifically for this setting with comparisons made between different SNA type studies to best understand what this study brings to the table.
2) I cannot confirm if the analysis is robust but commonly used R packages were utilised.
3) I think real time social network analysis software would be ideal but impossible at this time. Contact tracing investigation will still be manual unless we are all using Bluetooth/GPS data tracing. It simply isn’t feasible at this time in terms of the data load, privacy concerns and access to technology. I would urge the authors to come up with new recommendations based on their study.

Reviewer 3 ·

Basic reporting

The authors present a study on retrospective data of COVID19 occurrence and spread between January and August 2020. The authors outline their objectives and undertake this study using secondary data to elucidate on disease spread in Eastern Malaysia. Overall, the manuscript and rigor of analysis id good and well presented. Here are additional to improve the article.

Specific comments
In the third sentence in the abstract replacing ‘the’ in ‘the rural district’ with ‘a’ would be more appropriate grammatically since the reader has no idea what district is being referred to. Alternatively adding the name of the district would be better.
L48: Since the manuscript is in past tense, ‘is’ should be replaced with ‘was’.
L72: Consider citing R software too. ##
## To cite R in publications use:
##
## R Core Team (2021). R: A language and environment for statistical
## computing. R Foundation for Statistical Computing, Vienna, Austria.
## URL https://www.R-project.org/.
##
## A BibTeX entry for LaTeX users is
##
## @Manual{,
## title = {R: A Language and Environment for Statistical Computing},
## author = {{R Core Team}},
## organization = {R Foundation for Statistical Computing},
## address = {Vienna, Austria},
## year = {2021},
## url = {https://www.R-project.org/},
## }
L120: Replacing ‘have’ with ‘had’ would be more grammatically correct
L126 and elsewhere in the manuscript. Kindly check the sentence tense. For reported work, past tense is recommended.
L239-240: Indeed, a policy governing female caregivers would be a prudent thing but also awareness and education may go a long way in sensitizing the caregivers on the best practices. Consider adding this as a recommendation.

Experimental design

No comment

Validity of the findings

No comment

Additional comments

Specific comments
L67: Networks can be static or dynamic, directed or undirected, and/or weighted or unweighted. It would be good to inform the reader at this point, what type of networks were developed?
L85. OpenOrd is generally designed to handle undirected networks. In line 79, it is suggested that the networks were directed. Kindly provide more details about the type and nature of networks built and how the different algorithms were used.

---

## Round 0.2 · accepted · Accept

I agree with reviewer 1 that the manuscript is nearly ready for publication. Please follow their recommendations of the necessary edits for the final version.

Reviewer 3 ·

Basic reporting

L35-39: Since we know that nodes here refer to study subjects, would it not be better calling them people or individuals instead of nodes?
L76-77: Consider phrasing this ‘reacts to stop the illness’ as ‘responds to mitigate spread’
L147: In this phrase ‘how efficient a node transmit the infection’ consider adding ‘s’ at the end of ‘transmit’ for correctness
L148: This sounds incomplete ‘Influential in this study referred to the node with a high degree and high betweenness centrality value.’. Should there be a word between influential and in? E.g., Influential nodes….
L238: When we use ‘more prevalent’ the assumption is that we are comparing items which is not the case here.
L313-314: Although fomite transmission is possible, the survivability of the virus in the environment has been debated. It would be good to mention that too here so the reader can assess the risk on their own.
L315: This phrase could be presented differently for better sentence structure ‘of infection was able to be contained’ by removing the words ‘able to be’

Experimental design

No comment

Validity of the findings

No comment

Additional comments

The revised version of the manuscript reads superbly save for some minor edits I proposed above.